# Predicting Tumor Recurrence with Early 18F-FDG PET-CT After Thermal and Non-Thermal Ablation

**DOI:** 10.3390/curroncol32090521

**Published:** 2025-09-18

**Authors:** Govindarajan Narayanan, Nicole T. Gentile, Brian J. Schiro, Ripal T. Gandhi, Constantino S. Peña, Susan van der Lei, Madelon Dijkstra

**Affiliations:** 1Department of Interventional Radiology, Herbert Wertheim College of Medicine, Florida International University, Miami, FL 33199, USA; 2Department of Interventional Oncology, Miami Cancer Institute, Baptist Health South Florida, Miami, FL 33176, USA; 3Department of Interventional Radiology, Miami Cardiac and Vascular Institute, Baptist Health South Florida, Miami, FL 33176, USA; 4Department of Radiology and Nuclear Medicine, Amsterdam UMC, Location VUmc, Cancer Center Amsterdam, 1081 HV Amsterdam, The Netherlands

**Keywords:** 18-fluorodeoxyglucose (18F-FDG) positron emission tomography–computed tomography (PET-CT), microwave ablation (MWA), cryoablation, irreversible electroporation (IRE)

## Abstract

This study investigates positron emission tomography–computed tomography (PET-CT) performed within 24 h after image-guided tumor ablation to predict treatment success. Tumor ablation is a minimally invasive method that destroys cancer cells using heat, freezing, or electrical pulses and is applied to both primary and metastatic tumors. This study analyzed patients to determine if early PET-CT findings could identify residual cancer tissue. Visible areas of persistent tumor activity strongly predicted local tumor regrowth during follow-up. These results suggest that early PET-CT imaging can help detect incomplete tumor destruction, allowing prompt retreatment and potentially improving patient outcomes. Incorporating early PET-CT scans within 24 h after ablation could refine post-ablation assessment, guide clinical decision-making, and contribute to better long-term local cancer control.

## 1. Introduction

Ablative therapies are an effective, minimally invasive approach for treating tumors across the body in both primary and metastatic settings [1]. These include microwave ablation (MWA), cryoablation, and irreversible electroporation (IRE), which destroy cancerous tissue by different mechanisms. Thermal ablative modalities (MWA and cryoablation) cause cell death by coagulative necrosis, whereas IRE disrupts the homeostasis mechanism leading to apoptosis [2,3,4]. Failure to completely ablate cancerous cells can result in the development of local tumor recurrence, which is one of the main concerns following ablative treatments [1]. Complete ablation success is determined by the lack of enhancement after ablation on CT-guided ablation, and the size of the ablated zone is compared with the pre-ablation dimensions to determine the size of the tumor-free margin [5,6].

While this helps in hypervascularized tumors, such as hepatocellular carcinoma (HCC) and hypervascular metastases, it is not ideal in metastatic colorectal cancer (mCRC) and pancreatic cancer [7,8]. Therefore, relying on contrast enhancement (ce) and anatomic imaging like computed tomography (CT) limits the ability to adequately assess treatment success and tumor response. Functional imaging using 18-fluorodeoxyglucose (18F-FDG) positron emission tomography (PET) is used to identify regions of increased cellular metabolism and is commonly used in mCRC, pancreatic cancer, and lung cancer [9]. As areas of tumor growth and inflammation both show high 18F-FDG avidity, performing 18F-FDG PET-CT scans within 24 h post-ablation allows assessment of treatment success prior to the post-ablation inflammatory response [7]. Post-ablative inflammatory processes can continue to show 18F-FDG avidity for over 2–3 months, inhibiting tumor growth. Therefore, the post-ablation PET scan serves two purposes: (1) it gives an immediate assessment of the ablation zone, and (2) it serves as a baseline for follow-up PET scans 3 months post-ablation.

Studies that have investigated the ability of 18F-FDG PET-CT scans taken within 24 h of ablation to predict ablation success and tumor recurrence have been limited to RFA, with the exception of early 18F-FDG PET-CT scans post-IRE in pancreatic tumors [7,10]. Although these studies have shown positive results, they have not verified the diagnostic and predictive power of post-ablation PET for different ablative modalities and malignancies. The aim of this study was to retrospectively analyze the ablation effectiveness and the incidence and pattern of tumor recurrence after ablation for correspondence with 18F-FDG avidity in PET-CT scans taken within 24 h post-ablation in multiple cancer types over a series of ablation modalities.

## 2. Materials and Methods

This single-center retrospective study was administered in accordance with the ‘Strengthening the Reporting of Observational studies in Epidemiology’ (STROBE) guideline [11]. The affiliated Institutional Review Board granted permission for this study.

### 2.1. Patient Selection and Data Collection

Patients who underwent MWA, cryoablation, or IRE between August 2018 and February 2024 for primary and metastatic cancers were included. Baseline characteristics concerning patient, procedure, and tumor data were identified and collected from electronic medical records. These included sex, age, cancer type, tumor location, ablation modality, and tumor size. Patients without at least one month of follow-up and patients who received immediate re-treatment were excluded.

### 2.2. Ablation Procedures

All procedures were performed percutaneously with needle probe placement under CT guidance, (ultrasound) US guidance, and/or robotic navigation (Epione, Quantum surgical, Montpellier, France) under general anesthesia [12,13]. Before the procedure, a ceCT scan was conducted to ensure the patient’s optimal positioning and aid in accessing the target lesion. All procedures were performed in accordance with the instructions for use provided by the manufacturer and quality guidelines [14].

Emprint^TM^ (Medtronic-Covidien, Minneapolis, MN, USA), NeuWave^TM^ (Johnson and Johnson, Madison, WI, USA), or Solero device (AngioDynamics, Queensbury, New York, NY, USA) generators with compatible antennas were used for the MWA procedures. Microwave ablation uses dielectric hysteresis to create heat, which destroys tissue by heating it to deadly temperatures using an electromagnetic field typically between 900 and 2500 MHz.

For cryoablations, the ICEFX™ system (Boston Scientific, Marlborough, MA, USA) was used. Cryoablation, which involves freezing to destruct tissues therapeutically, triggers cell membrane rupture, dehydration, and local tissue ischemia, resulting in complete tissue destruction within temperature ranges of −20 to −50 °C. The typical procedure comprises a freeze cycle followed by a passive thaw cycle to optimize cell death.

IRE procedures were performed using the NanoKnife^TM^ device (AngioDynamics, Queensbury, New York, NY, USA). Using a configuration that produces high-voltage direct current pulses (ranging from 1500 to 3000 V) at currents of 25 to 45 A, typically, 70 pulses were administered in seven sets, each containing 10 pulses, between paired unipolar electrodes. The voltage for each electroporation was determined based on the distance between electrodes, aiming to reach between 1000 V and a maximum of 1500 V.

Following the ablation procedure, a ceCT scan was performed to ensure complete ablation of the target area, evaluate blood vessels, and check for any immediate post-procedural complications. Patients were then extubated and transferred to the recovery area for at least 4 h, followed by admission for overnight observation. The next day, patients had received a PET scan in the morning and clinical examinations and routine laboratory tests, such as a complete blood count and serum chemistry profile. Discharge was arranged once stability and a low risk of post-procedural infections or complications were confirmed.

### 2.3. Imaging and Follow-Up

The study analyzed the post-ablation 18F-FDG PET-CT scans conducted within 24 h of the procedure. This was correlated with cross-sectional imaging, including CT and magnetic resonance imaging (MRI) at 1 month and every 3–4 months thereafter and/or 18F-FDG PET-CT scans every 3 months and as clinically required thereafter, which were used for follow-up assessments approximately every 3–4 months in the first year following ablation thereafter as clinically indicated.

### 2.4. Outcome Measures and Statistical Analysis

The primary outcome measure was the correlation of post-ablation 18F-FDG PET-CT findings with the development of local tumor progression (LTP) per tumor. The 18F-FDG PET-CT imaging was reported and/or reviewed by a board certified radiologist. When total photophenia and no 18F-FDG uptake were present, it was assumed that no viable tumor was present and the treatment was considered successful. Residual 18F-FDG uptake was divided into three subgroups and scored accordingly: rim-shaped 18F-FDG uptake, along the periphery of the ablation zone (partial or complete), focal 18F-FDG uptake and nodular 18F-FDG uptake. LTP was described as a solid and unequivocally enlarging mass, enhancement at diffusion-weighted MRI, or as focal 18F-FDG avidity at the surface of the ablated tumor. The secondary outcome measure was local tumor progression-free survival (LTPFS) per tumor.

Baseline patient, procedural, and tumor characteristics were described as numbers with percentages if categorical, as mean or median with range if continuous, and displayed in a table. The correlation between different subgroups of 18-FDG avidity was measured by the chi-square test. Significant p-values were set at <0.05. LTPFS was evaluated using the Kaplan–Meier survival curves. Furthermore, LTPFS was evaluated utilizing Cox proportional hazard regression models, which were adjusted for potential confounding factors in multivariable analysis. Initially, potential confounders were identified in univariable analysis (*p* < 0.100) and utilizing a backward selection procedure in multivariable analysis. Variables were considered potential confounders if their p-value was less than 0.050 in the final model. Actual confounders were identified when the regression coefficient in the Cox regression model for LTPFS changed by more than 10% in the corrected model. Hazard ratios (HR) and 95% confidence intervals (95% CI) were computed accordingly.

Statistical analyses were conducted using Excel (Microsoft) SPSS^®^ Version 28.0 (IBM^®^, Armonk, New York, NY, USA), and R version 4.2.1 (R Foundation, Vienna, Austria) [15,16].

## 3. Results

A total of 153 patients who had a 18F-FDG PET-CT scans within 24 h after MWA, cryoablation, or IRE between August 2018 and February 2024 were identified from electronic medical records (Figure 1). Twenty-four patients were excluded due to lack of follow-up (*n* = 20), immediate re-treatment (*n* = 3), and different PET-CT tracer (*n* = 1). A total of 132 patients, undergoing 159 procedures for 224 tumors, were included in the analyses.

### 3.1. Baseline Characteristics

Patient, procedure, and tumor characteristics are presented in Table 1. In this cohort, 55 patients were male (41.7%), and 77 patients were female (58.3%). The mean age of this cohort was 65 years (range 32–92). Most patients had colorectal and anal cancer (46.2%). The median number of procedures per patient was 1 (range 1–4). The median number of tumors per procedure was 1 (range 1–5). Most tumors were metastases (88.8%), located in the liver (69.6%). MWA was used in 126 tumors (56.3%), cryoablation in 9 tumors (4.0%), and IRE in 89 tumors (39.7%). Median tumor size was 18 mm (range 5–92).

### 3.2. Local Tumor Progression-Free Survival (LTPFS)

During follow-up, LTP developed in 120 out of 224 tumors (53.6%) (Figure 2). Altogether, 1-, 2-, and 3-year LTPFS was 44.2%, 30.8% and 26.8%, respectively, with median follow-up time of 9.7 months. Median time to LTP was 9.1 months (95% CI 6.3–11.9). After development of LTP, radiotherapy was administered for 7 tumors, surgical resection for 4 tumors, MWA for 17 tumors, MWA + transarterial chemoembolization (TACE) for 2 tumors, IRE for 11 tumors, embolization + IRE for 1 tumor, Y90 for 6 tumors. The rest of LTP patients received systemic therapy due to (extensive) disease progression, not locally retreated or were yet to undergo planned retreatment.

### 3.3. 18F-FDG PET-CT

The presence of 18F-FDG avidity on PET-CT within 24 h after the ablation was significantly correlated with development of LTP at follow-up imaging (*p* < 0.001; Table 2). Nodular 18F-FDG uptake was highest associated with development of LTP. The positive predictive value of nodular 18F-FDG avidity was 86.7%. The negative predictive value was 48.8%. Example cases are shown in Figure 3 and Figure 4.

### 3.4. Multivariable Analysis

In univariable analysis, 18F-FDG avidity was significantly associated with LTP with HR 2.203 (95% CI 1.515–3.204; *p* < 0.001). Univariable analysis identified two potential associations with LTP: sex (*p* = 0.011) and tumor size (*p* = 0.003) (Table 3). The variables were included in multivariable analysis to analyze potential confounders associated with 18F-FDG avidity influencing LTPFS. Sex (*p* = 0.003) was a significant confounder in multivariable analysis. Corrected HR for 18F-FDG avidity 2.355 (95% CI 1.614–2.647; *p* < 0.001).

### 3.5. Subgroup Analysis

A total of 53 patients who had a 18F-FDG PET-CT scans within 24 h after MWA of CRLM, between August 2018 and January 2025, undergoing 73 procedures for 102 tumors, were included in the analyses. In this cohort, 27 patients were male (50.9%) and 26 patients were female (49.1%). The mean age of this cohort was 63 years (range 38–92). The median number of procedures per patient was 1 (range 1–5). The median number of tumors per procedure was 2 (range 1–5). The median tumor size was 16 mm (range 5–92).

During follow-up, 1-year LTP occurred in 40 out of 102 tumors (39.2%). Table 1 shows a significant difference in 1-year LTP divided by no-avidity and avidity at 18-FDG PET-CT scans within 24 h (*p* = 0.015). Median follow-up time was 4.9 months. The sensitivity of 18F-FDG avidity was 30.0% (Table 4). The specificity was 90.3%. In univariable analysis, 18F-FDG avidity was significantly associated with LTP with an HR of 3.663 (95% CI 1.838–7.299, *p* < 0.001), as shown in Figure 5.

## 4. Discussion

In this retrospective analysis, the presence of 18F-FDG avidity on PET-CT within 24 h after the ablation was significantly correlated with the development of LTP at follow-up imaging. The positive predictive value of nodular 18F-FDG avidity was 86.7%. In multivariable analysis, 18F-FDG avidity was significantly related to LTPFS with an HR of 2.355 (95% CI 1.614–2.647).

Advancements in ablative techniques, alongside developments in image fusion and navigation systems, have led to improved tumor visualization and precise needle positioning. An essential determinant of local tumor control post-thermal ablation is the establishment of an adequate ablation margin, representing the distance from the initial lesion boundaries to the border of the post-treatment ablation zone [1,17]. Studies have demonstrated that circumferential safety margins of at least 5 mm, and ideally greater than 10 mm, substantially enhance local control [17,18,19]. Through the integration of image fusion and the prediction of ablation margins, technical success (A0 ablations) can be achieved, ensuring the attainment of crucial prognostic factors for LTP [17,18,20,21,22]. The cumulative effect of recent and ongoing enhancements has resulted in heightened local tumor control and increased LTPFS rates. Another technique that has also shown potential for enhanced local disease control is improving tumor visibility during thermal ablation [23,24]. CT hepatic arteriography (CTHA) involves administering a contrast agent via a catheter inserted into the hepatic artery. CTHA also aids in detecting additional tumors or local tumor progression at the edge of prior ablation zones, also known as the ‘incomplete ring sign’ [25,26].

To enable local retreatments aimed at achieving a cure, it is essential to promptly detect instances of residual local disease. A ceCT scan is often used to assess the ablation zone and possible residual disease. However, a study by Cornelis et al. failed to significantly correlate ceCT with local recurrence [27]. The challenging differentiation between post-ablation inflammation effects and residual viable tumor tissue makes it difficult to draw conclusions from ceCT. Additionally, the ability of ceCT or MRI to immediately distinguish between ablation cavity hemorrhage and residual tumor is limited [28]. In the current literature, while studies have demonstrated positive predictive outcomes of 18F-FDG avidity, there remains a lack of verification regarding the diagnostic and predictive efficacy of this technique across various ablative modalities and malignancies in larger cohorts. Since both tumor growth and inflammation exhibit heightened 18F-FDG avidity, obtaining 18F-FDG PET-CT scans within 24 h post-ablation enables the evaluation of treatment efficacy before the onset of the post-ablation inflammatory response [7]. The inflammatory processes following ablation can sustain 18F-FDG avidity for up to 2–3 months, impeding tumor growth.

Previous studies examining the predictive capability of 18F-FDG PET-CT scans taken within 24 h of ablation have been predominantly limited to RFA. Liu et al. included twelve patients with twenty metastases undergoing PET-CT within 24 h after RFA. This study showed three metastases with mild rim-shaped 18F-FDG uptake, with two biopsy-proven residual metastases and one disappearance of 18F-FDG uptake after 6 months, referred to as inflammatory [7]. Eleven patients were evaluated for residual tumor after RFA by Veit et al., and they presented a sensitivity of 65% for PET-CT [29]. Another study only evaluated 25 liver metastases from different origins treated by RFA and MWA and found a high accuracy of 92% of 18F-FDG PET-CT, yet no significant correlation of 18F-FDG PET-CT and LTP at 1-year follow-up. By evaluating 21 patients treated with RFA and MWA, Cornelis et al. demonstrated the superiority of immediate 18F-FDG PET-CT over ceCT in predicting treatment success [27]. For the detection of residual viable tumor cells, 18F-FDG PET-CT had a sensitivity of 66.7% and specificity of 95% in a study of 26 lesions treated with cryoablation and RFA [30]. A possible explanation of the relatively lower rates of sensitivity may be related to the susceptibility of 18F-FDG uptake of inflammatory processes [7]. Additionally, the 24 h time interval might be too wide, and the limitation to 12 h may be sufficient, according to a study by Veit et al. [29]. A similar retrospective study by Vandenbroucke et al., which included 20 patients, showed that nodular 18F-FDG uptake was highly predictive for LTP with a positive predictive value of 100% [31]. Following IRE in pancreatic tumors, low levels of SUV at immediate 18F-FDG PET-CT were associated with higher overall survival [10]. The correlation between the predictive role of 18F-FDG PET-CT within 24 h after ablation and the first follow-up substantiates close follow-up for 18-FDG avid tumors. Zadech et al. reported on the additive value of PET–CT-guided ablation; however, the costs and the infrastructure will not make this possible for most institutions [32].

The considerable quantity of tumors included in the study provided sufficient statistical power, thereby enhancing its validity. However, the absence of randomization in the study design represents a significant drawback, possibly leading to selection bias and confounding variables. Despite conducting further multivariable analysis to mitigate potential confounders, complete elimination of residual confounding cannot be guaranteed. As LTP was calculated per tumor, this may lead to intra-patient correlation and may underestimate variance and inflate type 1 error. Several ablation modalities were included in this study, not showing an association with LTPFS in univariable analysis. The study’s retrospective single-center design further adds to its limitations, potentially impacting the generalizability of findings and introducing inherent biases. Furthermore, the high number of different cancer types might not contribute to the distinction of results for each type of cancer; this heterogeneity may lead to spectrum bias, and the diversity of tumor types may have influenced LTP. Subgroup analyses are performed for the largest group to compare results. Given the lack of a standard follow-up protocol after ablation in vascular tumors, this study is a starting point to further assess the role of post-ablation PET-CT as a tool to predict LTP. Confirmation software are available; however, they are not yet validated in large clinical trials. Confirmation ablation software was not routinely used, though it is currently utilized in all cases, and it is hypothesized that implementation of this measure will decrease LTP. Additionally, LTP was defined radiologically and not systematically confirmed by biopsy; this may lead to misclassification.

## 5. Conclusions

To conclude, the presence of 18F-FDG avidity on PET-CT within 24 h after the ablation was highly correlated with the development of LTP and decreased LTPFS. The detection of residual tumor tissue may allow early retreatment, especially in tumors with nodular uptake after ablation, contributing to increased local disease-free survival.

## Figures and Tables

**Figure 1 curroncol-32-00521-f001:**
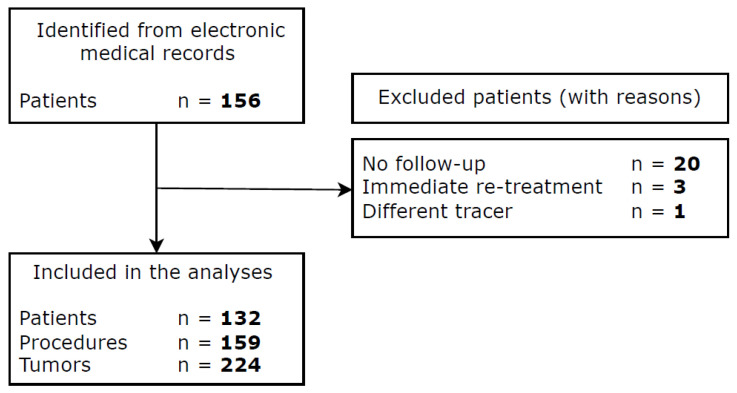
Flowchart of included patients.

**Figure 2 curroncol-32-00521-f002:**
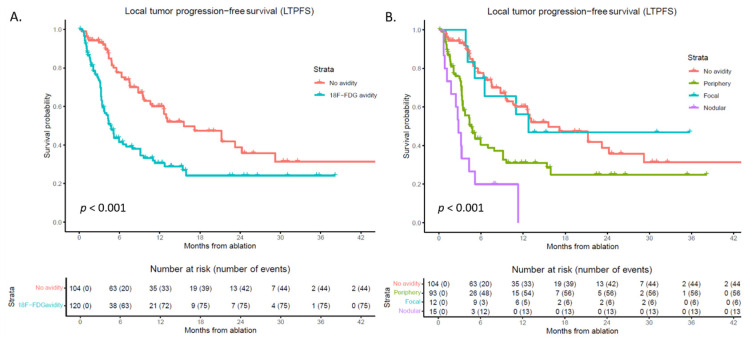
Kaplan–Meier curves of local tumor progression-free survival (LTPFS) per tumor after ablation. Numbers at risk (number of events) are per tumor. (**A**) No avidity (pink), 18F-FDG avidity (blue). Overall comparison log-rank (Mantel–Cox) test, *p* < 0.001 (**B**) No avidity (pink), periphery (green), focal (blue), nodular (purple). Overall comparison log-rank (Mantel–Cox) test, *p* < 0.001.

**Figure 3 curroncol-32-00521-f003:**
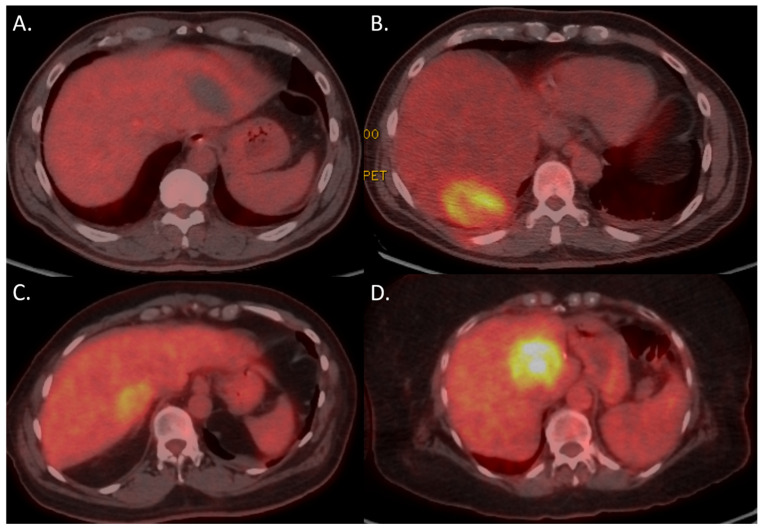
18F-FDG PET-CT within 24 h in four different patients with (**A**) no 18F-FDG uptake; (**B**) (partially) rim-shaped 18F-FDG uptake along the periphery of the ablation zone; (**C**) focal 18F-FDG uptake; and (**D**) persistent nodular 18F-FDG uptake.

**Figure 4 curroncol-32-00521-f004:**
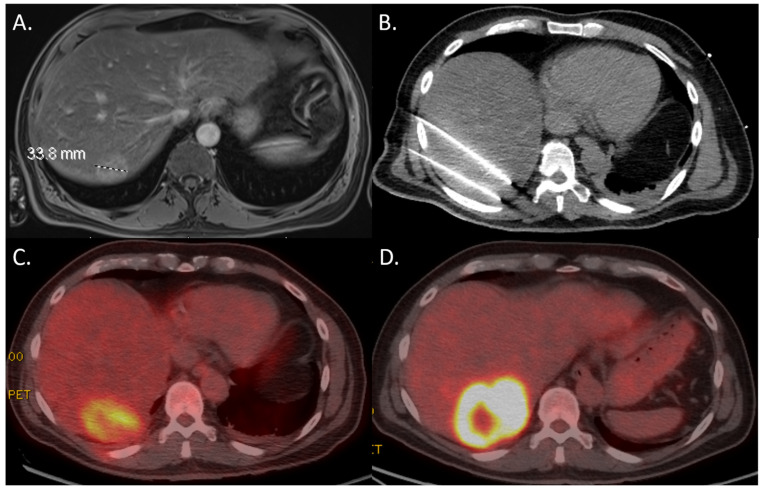
A 57-year-old male with pancreatic cancer with a liver metastasis in segment 7. (**A**) Pre-procedural MRI demonstrating intermediate T2 signal and rim enhancement consistent with a metastasis measuring 34 mm. (**B**) Intra-procedural CT-guided 17-gauge NanoKnife IRE ablation needle placement (2 out of 4 needles). (**C**) 18F-FDG PET-CT scan within 24 h after IRE demonstrating (partially) rim shaped 18F-FDG uptake along the periphery of the ablation zone. (**D**) Follow-up 18F-FDG PET-CT scan after 3 months showing thick peripheral FDG uptake located at the ablation cavity consistent with progression of residual disease.

**Figure 5 curroncol-32-00521-f005:**
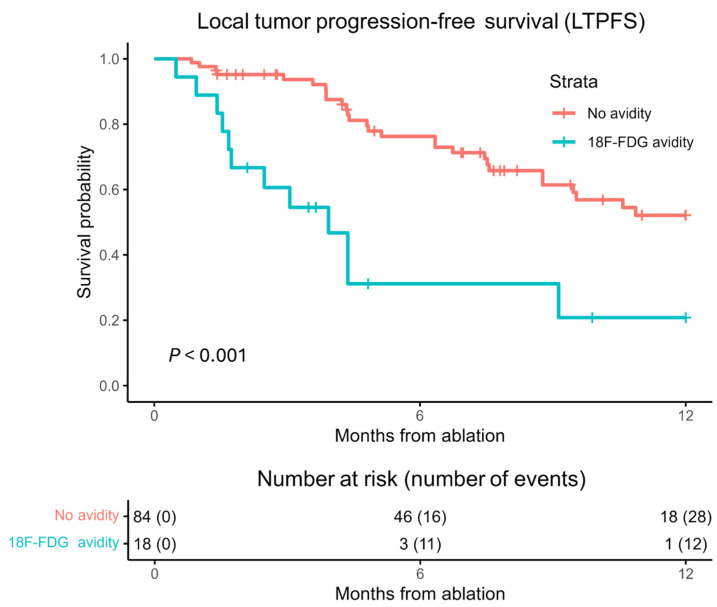
Kaplan–Meier curves of 1-year local tumor progression-free survival (LTPFS) per tumor after MWA. Numbers at risk (number of events) are per tumor. No avidity (pink), 18F-FDG avidity (blue). Overall comparison log-rank (Mantel–Cox) test, *p* < 0.001.

**Table 1 curroncol-32-00521-t001:** Patient, procedural, and tumor characteristics.

Patient Characteristics		Value (N = 132)
Sex	Male	55 (41.7%)
	Female	77 (58.3%)
Age in years (mean, range)		65 (32–92)
Primary cancer origin *	CRC and anal	61 (46.2%)
	HCC	2 (1.5%)
	Pancreatic	18 (13.6%)
	Biliary	5 (3.8%)
	Breast	12 (9.1%)
	Lung	9 (6.8%)
	RCC and urothelial	13 (9.8%)
	Gynecological	4 (3.0%)
	Other	9 (6.8%)
Number of procedures per patient	1	111 (84.1%)
	2	16 (12.1%)
	3	4 (3.0%)
	4	1 (0.8%)
**Procedure characteristics**		**Value (N = 159)**
Number of tumors per procedure	1	113 (71.1%)
	2	31 (19.5%)
	3	12 (7.5%)
	4	2 (1.3%)
	5	1 (0.6%)
**Tumor characteristics**		**Value (N = 224)**
Tumor setting	Primary	25 (11.2%)
	Metastatic	199 (88.8%)
Tumor location	Lymph node	15 (6.7%)
	Liver	156 (69.6%)
	Lung	9 (4.0%)
	Pancreas	10 (4.5%)
	Kidney or adrenal	10 (4.5%)
	Bone	2 (0.9%)
	Soft tissue	22 (9.8%)
Ablation modality	MWA	126 (56.3%)
	Cryoablation	9 (4.0%)
	IRE	89 (39.7%)
Tumor size in mm (median, range)		18 (5–92)

Categorical variables are reported as number (%) of patients, continuous variables are reported as mean or median (range), * = 1 patient had two types of cancer, CRC = colorectal, HCC = hepatocellular carcinoma, RCC = renal cell carcinoma.

**Table 2 curroncol-32-00521-t002:** Correlation between 18F-FDG avidity on PET-CT within 24 h after the ablation and development of local tumor progression (LTP) during follow-up.

Local Tumor Progression	Yes	No	*p*-Value
	N = 120	N = 104	
18F-FDG avidity			
Nodular	13 (10.8%)	2 (1.9%)	
Focal	6 (5.0%)	6 (5.8%)	
Periphery	56 (46.7%)	37 (35.6%)	
No	45 (37.5%)	59 (56.7%)	0.005 *

Values are reported as number (%) of patients, * = Pearson Chi-Square.

**Table 3 curroncol-32-00521-t003:** Univariable and multivariable Cox regression analysis to detect variables associated with local tumor progression-free survival (LTPFS).

		Univariable Analysis		Multivariable Analysis	
		HR (95% CI)	*p*-Value	HR (95% CI)	*p*-Value
18F-FDG avidity	No	Reference	**<0.001**	Reference	**<0.001**
	Yes	2.203 (1.515–3.204)		2.355 (1.614–3.436)	
**Patient-related characteristics**					
Sex	Male	Reference	**0.011**	Reference	**0.003**
	Female	1.638 (1.120–2.396)		1.805 (1.230–2.647)	
Age		0.997 (0.984–1.010)	0.663		
Primary cancer origin	CRC and anal	Reference	0.683		
	HCC	1.362 (0.332–5.585)			
	Pancreatic	1.841 (1.009–3.359)			
	Biliary	1.101 (0.433–2.797)			
	Breast	1.176 (0.633–2.187)			
	Lung	1.411 (0.565–3.521)			
	RCC and urothelial	1.518 (0.834–2.764)			
	Gynecological	1.750 (0.425–7.198)			
	Other	1.022 (0.441–2.365)			
Number of procedures per patient	1	Reference	0.290		
	2	1.118 (0.690–1.813)			
	3	1.140 (0.655–1.983)			
	4	2.679 (0.970–7.400)			
**Procedure-related characteristics**					
Number of tumors per procedure	1	Reference	0.239		
	2	1.169 (0.780–1.753)			
	3	0.589 (0.339–1.024)			
	4	0.812 (0.324–2.033)			
	5	NA			
**Tumor-related characteristics**					
Tumor setting	Primary	Reference	0.359		
	Metastatic	0.781 (0.461–1.324)			
Tumor location	Lymph node	Reference	0.312		
	Liver	0.902 (0.472–1.752)			
	Lung	1.414 (0.517–3.872)			
	Pancreas	1.958 (0.778–4.930)			
	Kidney or adrenal	0.397 (0.087–1.808)			
	Bone	1.021 (0.131–7.982)			
	Soft tissue	1.178 (0.501–2.768)			
Ablation modality	MWA	Reference	0.259		
	Cryoablation	0.808 (0.294–2.219)			
	IRE	1.327 (0.918–1.919)			
Tumor size		1.017 (1.006–1.029)	**0.003**	1.011 (0.998–1.024)	0.099

HR = hazard ratio, 95% CI = 95% confidence interval, NA = insufficient group comparison, MWA = microwave ablation, IRE = irreversible electroporation. Using backward selection procedure, results of step by step removed variables were reported. Results are from last step of removal.

**Table 4 curroncol-32-00521-t004:** Sensitivity and specificity of LTP at 1 year divided by PET-avidity and no avidity.

	LTP	No LTP
PET-avidity	12 (sensitivity 30.0%)	6
No avidity	28	56 (specificity 90.3%)

## Data Availability

The data presented in this study are available on request from the corresponding author.

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
