# Peer review of "Predicting Tumor Recurrence with Early 18F-FDG PET-CT After Thermal and Non-Thermal Ablation"

_curroncol, 2025, doi:10.3390/curroncol32090521_

Round 1

Reviewer 1 Report

Comments and Suggestions for Authors

The authors report a single-center retrospective series (132 patients, 224 tumors) evaluating whether 18F-FDG PET-CT performed within 24 hours after percutaneous ablation predicts local tumor progression (LTP). They find that nodular FDG uptake within 24 h strongly correlates with later LTP and that PET avidity is associated with worse LTP-free survival.

Some considerations:

  1. The cohort includes a large mix of primary tumour types (CRC most common, but also HCC, pancreatic, breast, lung, RCC, etc.; see Table 1), and FDG-affinity varies markedly by histology and differentiation. For example, well-differentiated HCC is often FDG-negative (and may be choline-positive), and many luminal (A/B) breast cancer metastases have low FDG uptake. This heterogeneity can cause spectrum bias (false negatives in inherently low-FDG lesions and differing PPV/NPV across subgroups). The limitation is acknowledged in parts of the discussion but must be clearly stated as a major limitation.
  2. The endpoint “local tumor progression” is defined radiologically and not systematically confirmed by biopsy. Imaging-based definitions can misclassify inflammation, hemorrhage, or fibrosis as progression or vice versa. Please elaborate.
  3. Several patients had >1 tumor treated (median procedures per patient = 1 but range up to 4; tumors per procedure up to 5). The current Cox / Kaplan-Meier analyses appear to treat tumors as independent observations. This ignores intra-patient correlation and may underestimate variance and inflate type I error.
  4. The series mixes MWA, cryoablation (only 9 tumors), and IRE (89 tumors). Device types, operator technique, and thermal vs non-thermal mechanisms differ and may affect residual uptake and recurrence patterns.

Reviewer 2 Report

Comments and Suggestions for Authors

Dear editor in chief,

Thank you for giving me the opportunity to review for your journal.

This is a well-written, clinically relevant study on the utility of early (within 24 hours) 18F-FDG PET-CT imaging in predicting local tumor progression (LTP) following percutaneous ablation of primary and metastatic tumors. The method is OK and results clearly were used for conclusion making.

Major issues:

1- Including different tumors in the study although increased the sample size, it could increase the heterogeneity and should be addressed with subgroup analyses which on its own poses the problem of sample size in each group. This is a major limitation of the study.

2- What is the rational of classification of uptakes into rim, focal, and nodular patterns?

3- The statistical analyses for censored data is very well performed and the authors should be commended in this regard. 

4- The high positive predictive value (86.7%) for nodular uptake can be clinically important. The authors may discuss how this finding can be useful for possible early re-treatment.

5- Consider mentioning why 24 hours was chosen as the optimal time for imaging?

6- Explain more regarding low negative predictive value and its importance.

7- Please provide a table for comparison of previous studies on the post ablation FDG uptake for comparison of their results with the current study.

Best wishes

Round 2

Reviewer 1 Report

Comments and Suggestions for Authors

The authors properly revised the manuscript. 

Reviewer 2 Report

Comments and Suggestions for Authors

Dear editor,

The authors have addressed the issues brought up by the reviewers.